# Molecular basis of ubiquitin-specific protease 8 autoinhibition by the WW-like domain

Keijun Kakihara [1,2], Kengo Asamizu[1], Kei Moritsugu[3], Masahide Kubo[1], Tetsuya Kitaguchi[4], Akinori Endo[2], Akinori Kidera[3], Mitsunori Ikeguchi [3], Akira Kato[1], Masayuki Komada [1,2,5✉] & Toshiaki Fukushima [1,2,5✉]

Ubiquitin-specific protease 8 (USP8) is a deubiquitinating enzyme involved in multiple membrane trafficking pathways. The enzyme activity is inhibited by binding to 14-3-3 proteins. Mutations in the 14-3-3-binding motif in USP8 are related to Cushing's disease. However, the molecular basis of USP8 activity regulation remains unclear. This study identified amino acids 645–684 of USP8 as an autoinhibitory region, which might interact with the catalytic USP domain, as per the results of pull-down and single-molecule FRET assays performed in this study. In silico modelling indicated that the region forms a WW-like domain structure, plugs the catalytic cleft, and narrows the entrance to the ubiquitin-binding pocket. Furthermore, 14-3-3 inhibited USP8 activity partly by enhancing the interaction between the WW-like and USP domains. These findings provide the molecular basis of USP8 autoinhibition via the WW-like domain. Moreover, they suggest that the release of autoinhibition may underlie Cushing's disease due to USP8 mutations.

[1] School of Life Science and Technology, Tokyo Institute of Technology, Yokohama, Japan. [2] Cell Biology Center, Institute of Innovative Research, Tokyo Institute of Technology, Yokohama, Japan. [3] Graduate School of Medical Life Science, Yokohama City University, Yokohama, Japan. [4] Laboratory for Chemistry and Life Science, Institute of Innovative Research, Tokyo Institute of Technology, Yokohama, Japan. [5] These authors jointly supervised this work: Masayuki Komada, Toshiaki Fukushima. ✉email: makomada@bio.titech.ac.jp; tofu@bio.titech.ac.jp

Ubiquitin-specific protease 8 (USP8), a member of the USP family of deubiquitinases, regulates multiple membrane trafficking pathways and membrane fusion/fission events. It deubiquitinates endosomal membrane proteins, including epidermal growth factor receptor (EGFR), which enhances their recycling to the plasma membrane and/or inhibits their lysosomal degradation[1–8]. USP8 also maintains the protein levels and/or functions of endocytosis machinery components, including Eps15, HRS, STAM1/2 and CHMP proteins[9–12]. In addition, USP8 regulates autophagy and mitophagy by deubiquitinating EPG5, p62 and Parkin[13–15]. It suppresses the endoplasmic reticulum export of procollagen by deubiquitinating COPII protein Sec31[16] and enables cytokinesis, probably by deubiquitinating VAMP8—a SNARE required for vesicle fusion in cytokinesis[17]. Given the several functions of USP8, the regulatory mechanism of its enzyme activity must be elucidated; however, limited mechanistic data have been reported till date.

Similar to other USPs, USP8 harbours a catalytic USP domain comprising three subdomains, the fingers, palm and thumb[18,19] (Fig. 1a). These subdomains cooperatively form the ubiquitin-binding pocket, which recognises the globular core of the distal ubiquitin molecule. A deep cleft between the palm and the thumb subdomains functions as a catalytic cleft at which the catalytic triad (Cys, His and Asp/Asn residues) attacks the isopeptide bond between the distal ubiquitin C-terminal tail and proximal ubiquitin (or substrate protein). The crystal structure of the USP8 catalytic domain in its apo form shows its unique features[20]. Two loops, namely, blocking loops 1 and 2 (BL1 and BL2), which are utilised for ubiquitin recognition in other USPs, are positioned in a closed conformation. Furthermore, the fingers subdomain is tightened inwardly, making the ubiquitin-binding pocket too narrow to capture ubiquitin. Despite these unfavourable features, the USP domain effectively catalyses the deubiquitination reaction[20], implying the occurrence of substrate-induced conformational changes.

Some USPs are activated by specific interacting proteins that bind to the catalytic[21–25] or other domains[26–28] of USPs. The binding of USP8, which has a phosphorylation-dependent 14-3-3-binding motif (Fig. 1b), to 14-3-3 protein inhibits its enzyme activity[29]. The binding mode exhibits the characteristic features of mode I of 14-3-3-binding (RXX[pS/pT]XP)[30]. In the mitotic M-phase, when USP8 positively regulates cytokinesis, it dissociates from 14-3-3 and shows increased enzyme activity[29]. On the other hand, in corticotroph adenomas of Cushing's disease, ~50% of tumour samples show somatic mutations of USP8 that cause deletion or substitution of amino acids around the 14-3-3-binding motif[31,32]. Cushing's disease begins with the hypersecretion of adrenocorticotropic hormone (ACTH) from corticotrophs, followed by excess adrenal glucocorticoid. In cultured corticotrophs, the exogenous expression of USP8 lacking functional 14-3-3-binding motifs effectively enhances ACTH secretion[31], suggesting that 14-3-3-binding motif dysfunction causes this disease. Thus, 14-3-3 plays important role in regulating USP8 under physiological and pathological conditions. However, the molecular basis of 14-3-3-dependent inhibition of USP8's enzyme activity remains unclear.

This study aimed to elucidate the regulatory mechanism of USP8 enzyme activity. We identified a WW-like domain in USP8 as an autoinhibitory region. This WW-like domain and ubiquitin were found to competitively bind to the ubiquitin-binding pocket of USP8. In addition, 14-3-3 inhibited the USP8 enzyme activity partly by enhancing the interaction between the WW-like and catalytic domains.

## Results

**USP8 has an autoinhibitory region.** We generated a series of USP8 deletion mutants (Fig. 1b and Supplementary Fig. 1a) and evaluated their deubiquitinating activities in vitro. A mutant lacking amino acids (aa) 414–714 (USP8$^{\Delta414-714}$) and USP8$^{\Delta605-714}$ (Supplementary Fig. 1b) deubiquitinated ubiquitin chains faster than the wild-type USP8 (USP8$^{WT}$) and other mutants, suggesting that aa 605–714 contains one or more autoinhibitory regions that decrease the deubiquitinating activity of USP8. Further mutational analyses determined aa 645–684 be a putative autoinhibitory region (Fig. 1c).

Next, we tested the labelling efficiency of USP8$^{WT}$ or USP8$^{\Delta645-684}$ with ubiquitin-vinyl methyl ester (Ub-VME), which can be irreversibly conjugated to the Cys residue of the catalytic triad[33]. Ub-VME was able to label USP8$^{\Delta645-684}$ more effectively than it labelled USP8$^{WT}$ (Fig. 1d), implying that aa 645–684 hinders ubiquitin recognition by the USP domain.

We then examined the effects of aa 645–684 deletion on the ubiquitination levels of USP8 substrates in cells. EGF-induced ubiquitination of EGFR, a representative USP8 substrate, was lower in HeLa cells stably expressing USP8$^{\Delta645-684}$ than in cells expressing USP8$^{WT}$ (Fig. 1e and Supplementary Fig. 1c). We also examined the ubiquitination of other USP8 substrates such as Parkin[15], STAM1[10] and STAM2[2] using HEK293T cells over-expressing USP8$^{WT}$ or USP8$^{\Delta645-684}$. The ubiquitination levels of these substrates were decreased by USP8$^{WT}$ overexpression but were decreased more strongly by USP8$^{\Delta645-684}$ overexpression (Supplementary Fig. 1d–f). These results suggest that aa 645–684 decreases USP8 deubiquitinating activity in cells. This region might not regulate the intra-cellular localisation of USP8 because USP8$^{\Delta645-684}$ showed a similar staining pattern to that of USP8$^{WT}$ (Supplementary Fig. 1g). Taking together with in vitro data (Fig. 1b–d), we concluded that aa 645–684 functions as an autoinhibitory region.

**Autoinhibitory region is classified as an atypical WW domain.** The amino acid sequence of the autoinhibitory region is well conserved among vertebrates (Fig. 2a) and shows sequence similarity to the MAGI1 WW domain (aa 359–392) (Fig. 2b). Typical WW domains have two conserved tryptophan residues; however, some WW domains in MAGI1, SAV1 and WWOX have only the first residue (Fig. 2b), although these atypical WW domains show conformations that are similar to those of typical WW domains[34]. The autoinhibitory region of USP8 has only the first tryptophan residue (Trp$^{655}$), meaning that it shares a common feature with atypical WW domains.

We also performed in silico homology modelling using the MAGI1 WW domain (PDB ID: 2YSE)[35] as a template. In this model, the overall structure of the autoinhibitory region of USP8 is similar to that of canonical WW domains: a three-stranded antiparallel β-sheet[34] (Fig. 2c, left panel). Similar to WW domains, the β-sheet surface contains several aromatic or hydrophobic side chains (Phe$^{658}$, Pro$^{661}$, Phe$^{666}$, Tyr$^{668}$, His$^{670}$ and His$^{677}$) (Fig. 2c, right panel), indicating its role in protein–protein interactions. In addition, the N-terminal region, where Leu$^{651}$ and Pro$^{652}$ make contact with Trp$^{655}$, forms a hook structure that is often found in WW domains. Molecular dynamics (MD) simulations showed the stability of the model structure. The results also revealed that the substitution of Trp$^{655}$ to Ser (W655S) abolishes one of three β-strands and destabilises the domain structure (Fig. 2d). Thus, the first conserved tryptophan residue might underpin the folding, similar to the process observed in WW domains. On the basis of the common features of WW domains, we designated this region a WW-like domain.

We also examined the deubiquitinating activity of USP8, in which Trp$^{655}$ is substituted to Ser (USP8$^{W655S}$). We found that USP8$^{W655S}$ shows higher activity than USP8$^{WT}$ (Fig. 2e and

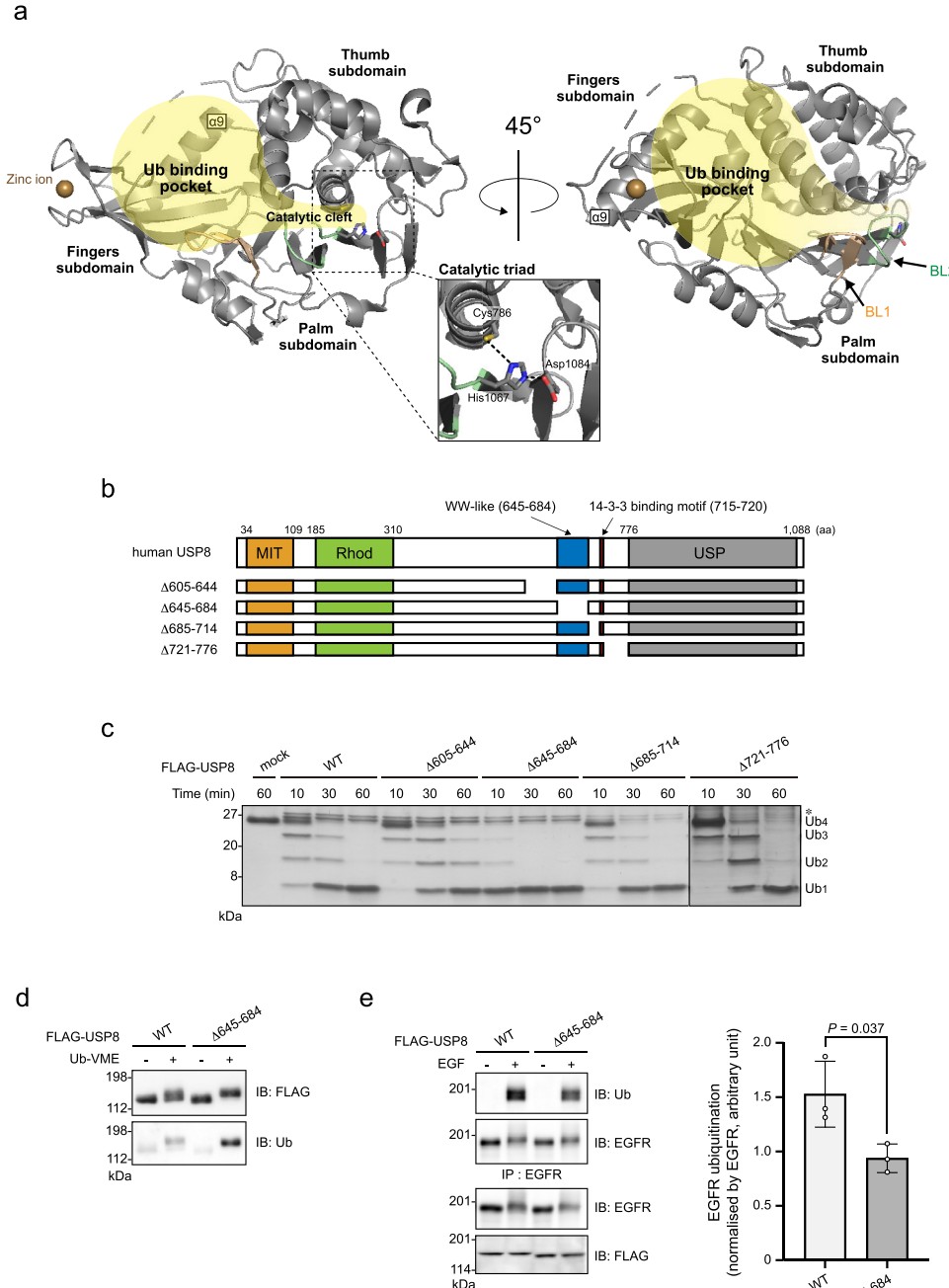

**Fig. 1 Effects of the deletion of amino acids (aa) 645–684 on USP8 activity. a** Crystal structure of the USP domain (aa 762–1109; PDB ID: 2GFO). Two representations are rotated by 45° around the Y-axis. The fingers, palm and thumb subdomains are indicated. The catalytic triad formed by Cys[786], His[1067] and Asp[1084] is shown in the enlarged view. The ubiquitin-binding pocket and catalytic cleft are indicated in yellow. The upper-left broken loop between Asp[888] and Asn[898] is non-structural. BL1, blocking loop 1; BL2, blocking loop 2. **b** Schematic structures of human USP8 and the truncated mutants used in **c**: MIT, MIT domain; Rhod, rhodanese-like domain; WW-like, WW-like domain identified in this study; USP, USP domain. **c** Deubiquitination activities of the USP8 mutants. HEK293T cells expressing FLAG-tagged USP8 mutants were lysed. Anti-FLAG immunoprecipitates were subjected to an in vitro deubiquitination assay using a Lys63-linked ubiquitin tetramer and then to SDS-PAGE and silver staining. *, copurified protein (s) with USP8. **d** Ubiquitin-vinyl methyl ester (Ub-VME)-labelling of USP8[Δ645-684]. Immunoprecipitated USP8 (WT or Δ645–684) were labelled and subjected to immunoblotting (IB). **e** EGFR ubiquitination levels in cells expressing USP8[Δ645-684]. HeLa cells stably expressing USP8 (WT or Δ645–684) were treated with EGF or left untreated. The lysates were subjected to immunoprecipitation (IP) and IB. The right graph shows EGFR ubiquitination levels normalised by EGFR in the immunoprecipitates (means ± standard deviations of three independent experiments). Statistical significance was determined using Student's *t*-test. See Supplementary Fig. 1.

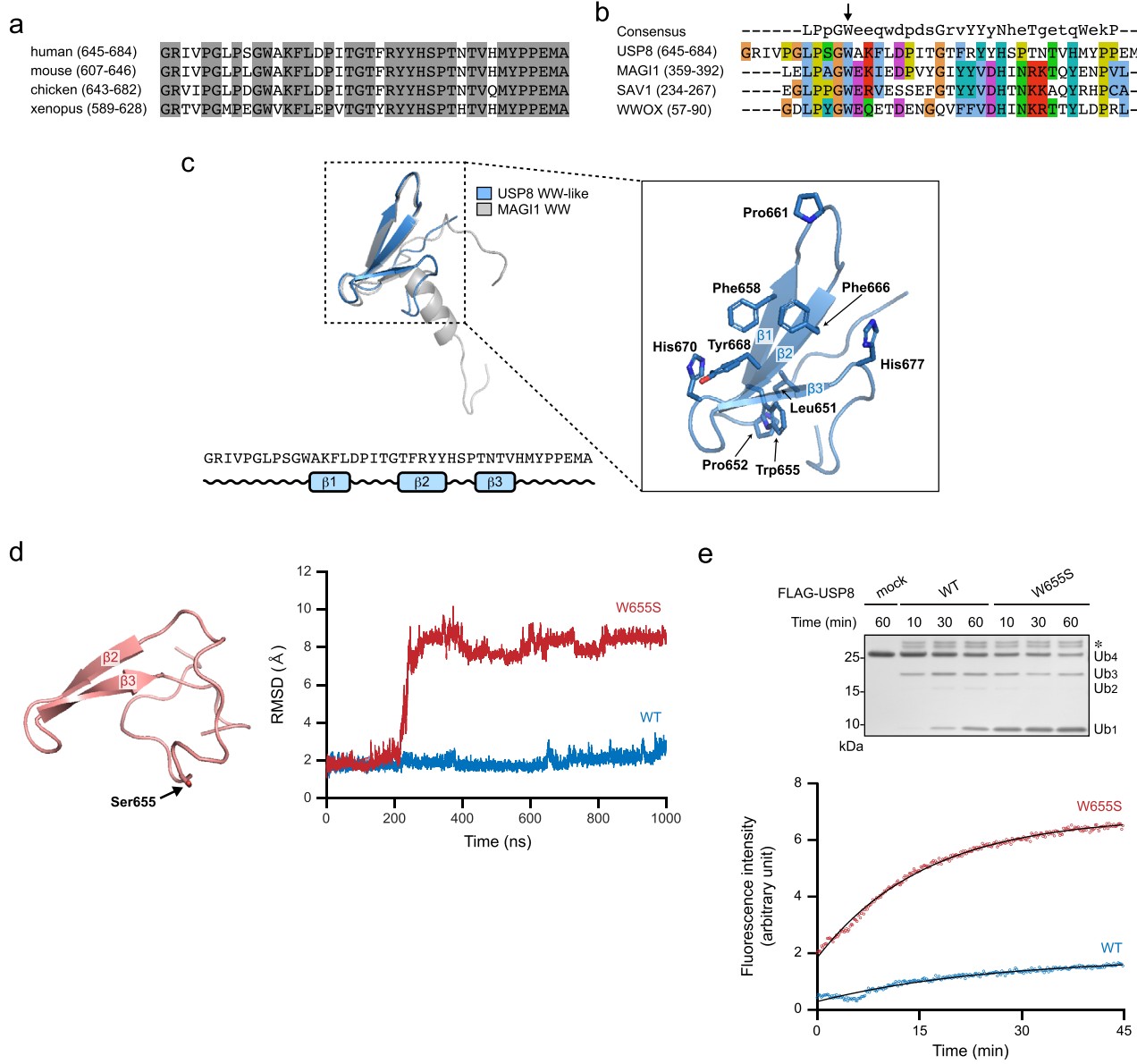

**Fig. 2 Structural modelling of the USP8 autoinhibitory region. a** Amino acid sequence conservation among vertebrates. Grey highlighting: conserved residues. **b** Sequence similarity to the atypical WW domains of MAGI1, SAV1 and WWOX. The arrow indicates the conserved N-terminal tryptophan. The consensus sequence of the WW domain is also shown. **c** A structural model of the WW-like domain. Left, superposition of the predicted model (blue) and MAGI1 WW domain (grey) as well as the secondary structure based on the model. Right, magnification of the hydrophobic surface formed by three antiparallel β-sheets. **d** Molecular dynamics simulations of the predicted model and Trp[655] to Ser (W655S) mutant. Left, the simulation structure of the W655S mutant. Right, the root mean square deviations (RMSD) from the predicted structure as a function of simulation time. Blue, WT. Red, W655S mutant. **e** Deubiquitination activities of USP8[W655S]. Top, ubiquitin chain cleavage assay by similar methods to those described in Fig. 1c. *, copurified protein (s) with USP8. Bottom, ubiquitin-AMC assay. HEK293 cells expressing USP8[W655S] were lysed. Anti-FLAG immunoprecipitates were subjected to an in vitro deubiquitination assay using ubiquitin-AMC. The fluorescence intensity of the released AMC was measured. See Supplementary Fig. 2.

Supplementary Fig. 2a, b). These results suggest that Trp[655] in the WW-like domain is important for its inhibitory function as well as folding.

**WW-like domain interacts with the catalytic USP domain**. We hypothesised that the WW-like domain interacts with the USP domain. A pull-down assay indicated the interaction of aa 1–714 of USP8 (USP8[1–714]) with the USP domain, whereas the deletion of the WW-like domain and the substitution of Trp[655] to Ser abolished the interaction (Fig. 3a, b and Supplementary Fig. 3a). These results indicate that the WW-like domain interacts with the USP domain in vitro.

We also performed a single-molecule fluorescence resonance energy transfer (FRET) assay (Fig. 3c) by constructing five FRET probes as shown in Fig. 3d. These probes comprise USP8[645–1118] (a region containing the WW-like and USP domains), EYFP and ECFP. Specifically, EYFP was fused to the N-terminus of the WW-like domain, whereas ECFP was inserted into five different surface loops of the USP domain (Fig. 3e). The insertion of ECFP in these probes is considered to not disrupt the tertiary structure of the USP domain because various USP family proteins have long insertion sequences at these surface loops[19]. When the WW-like and USP domains interact in these probes, a fluorescent signal from EYFP can be detected in response to the excitation of ECFP.

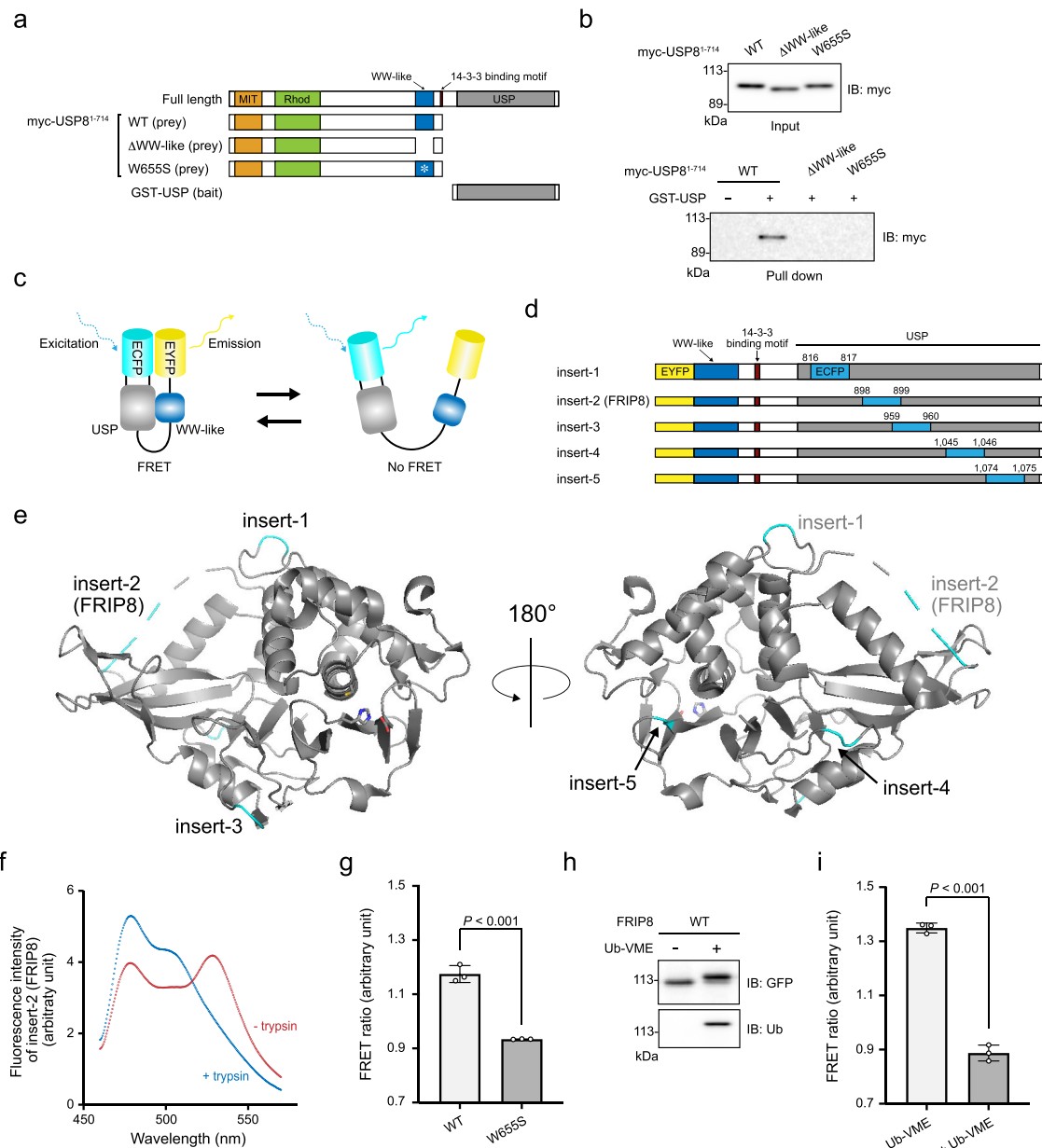

**Fig. 3 Interaction between the WW-like and USP domains. a** Schematic structures of the prey and bait proteins used in **b**. **b** Pull-down assay. Lysates of HEK293T cells expressing myc-tagged USP8$^{1-714}$ (WT or mutants) were incubated with GST-tagged USP domain, subsequently precipitated using glutathione beads. The adsorbed fraction and input were analysed using immunoblotting. **c** Schematic representation of single-molecule FRET probes used to detect the interaction. **d** Schematic structures of the probes. **e** ECFP insertion points are depicted on a crystal structure of the USP domain (PDB ID: 2GFO). Two representations are rotated by 180° around the Y-axis. **f** Emission spectra of FRIP8 at an excitation wavelength of 433 nm. Lysate of HEK293T cells expressing FRIP8 was subjected to fluorescence measurement with (blue) and without (red) trypsinisation. **g** FRET ratio of FRIP8$^{W655S}$. The emission spectra of FRIP8$^{W655S}$ were measured by similar methods to those described in **f**. The ratio of the emission intensity of EYFP to that of ECFP (i.e. the FRET ratio) was calculated. The graph shows the means ± standard deviations of three independent experiments. Statistical significance was determined using Student's *t*-test. **h**, **i** Ub-VME-labelling of FRIP8. Purified His-tagged FRIP8 was labelled with Ub-VME. It was subjected to immunoblotting (**h**) and a FRET assay (**i**). The graph shows the means ± standard deviations of three independent experiments. Statistical significance was determined using Student's *t*-test. See Supplementary Fig. 3.

We prepared lysates of cells expressing these probes and measured the emission spectra with the excitation wavelength for ECFP. As negative controls, we trypsinised the lysates to separate EYFP and ECFP in the probes, and then we measured their emission spectra. All probes showed significant FRET signals (Supplementary Fig. 3b). The expression of each probe and their trypsinisation were evaluated by immunoblotting (Supplementary Fig. 3c). FRET signals of insert-2 and insert-3 probes, in which the ECFP insertion sites are close to the ubiquitin-binding pocket

(Fig. 3e), were relatively higher. This implies that the pocket is involved in the efficient interaction with the WW-like domain. We named the insert-2 probe the FRET-based intramolecular interaction probe of USP8 (FRIP8), which is the name used hereafter. The detailed FRET spectra of FRIP8 are shown in Fig. 3f.

W655S substitution in FRIP8 significantly decreased the FRET signal (Fig. 3g and Supplementary Fig. 3d), indicating that the proper folding of the WW-like domain is important for the

interaction with the USP domain. We further purified His-tagged FRIP8 (Supplementary Fig. 3e) and treated it with Ub-VME, which was conjugated to FRIP8 (Fig. 3h) and could decrease the FRET signal (Fig. 3i), suggesting that the WW-like domain and ubiquitin molecule competitively interact with the ubiquitin-binding pocket.

A previous report indicated that USP8 dimerisation potentially occurs via the N-terminal MIT domain[20]. We found that USP8 dimerisation occurred via the co-immunoprecipitation of FLAG- and myc-tagged USP8s but the deletion of neither the WW-like nor the USP domain abolished this dimerisation (Supplementary Fig. 3f). Thus, these domains are apparently not required for USP8 dimerisation. Given that our FRET probes lacked the MIT domain, we speculate that the WW-like and USP domains interacted in an intramolecular manner in our FRET probes.

**WW-like domain plugs the USP domain's catalytic cleft and narrows the entrance to the ubiquitin-binding pocket.** In further assays, we analysed the interaction between the WW-like and USP domains in silico. First, we predicted the USP domain structure in ubiquitin-bound form by homology modelling using the ubiquitin-bound USP2 USP domain (PDB ID: 2HD5)[36] as a template. Unlike its apo form (PDB ID: 2GFO)[20], the predicted structure showed an open conformation in which the USP domain grabs the ubiquitin molecule with the tip of the fingers and the catalytic cleft (Fig. 4a and Supplementary Fig. 4a). Lys[913] and Leu[917] in an α9 helix (aa 902–918) (Fig. 1a) also make physical contact with ubiquitin. The C-terminal tail of ubiquitin is embedded in the catalytic cleft, supported by BL1 and BL2. The catalytic triad is aligned in the model structure.

We also attempted to identify the putative binding site (s) by examining the effects of point mutations in FRIP8 on the FRET signal. Results showed that substitution of Phe[1014] in the BL1 to Ala (F1014A) significantly decreased the FRET signal (Fig. 4b and Supplementary Fig. 4b). In addition, we examined the effects of F1014A mutation on the deubiquitinating activity of full-length USP8. Afterward, we measured the activity using a ubiquitin-AMC assay, which is more sensitive and quantitative than gel-based assays. The F1014A mutation enhanced the enzyme activity (Fig. 4c and Supplementary Fig. 4c). These findings suggest the involvement of Phe[1014] in the binding process.

Using structural data from the WW-like (Fig. 2c) and USP (Fig. 4a) domains, we performed in silico docking simulations. We selected the most probable structural model under the conditions that Phe[1014] is one of the binding sites and that the inter-domain interaction is the most favourable possibility (Supplementary Fig. 4d; see Methods section for more details). We found that this stable complex model includes a WW-like domain, which occupies part of the ubiquitin-binding pocket and plugs the catalytic cleft (Fig. 4d). An enlarged view of the contact sites is shown in Fig. 4e. Tyr[668] on the β-sheet of the WW-like domain forms a hydrophobic contact with Phe[1014] in the BL1. Thr[673] in the third β-strand of the WW-like domain makes contact with Leu[1063] in the BL2. Asp[660] and His[670] in the loops between β-strands in the WW-like domain form polar interactions with Lys[1011] and Gln[867] in the palm subdomain, respectively. BL1 and BL2 are likely relocated by these contacts from their positions in the ubiquitin-bound form (Fig. 4f).

Another important feature of this model is the slight bending of the fingers subdomain to the inside of the ubiquitin-binding pocket, which narrows the entrance to the pocket (Fig. 4f). The Pro[661] residue in the loop between the β-strands of the WW-like domain makes contact with the Pro[955] residue at the base of the fingers subdomain (Fig. 4e), which possibly contributes to the bending of the fingers. MD simulations confirmed that although the fingers subdomain moves flexibly in the apo form, it is fixed to this position in the WW-like domain-bound form (Fig. 4g).

We also performed mutation analysis targeting two putative binding sites: Pro[955] in the fingers subdomain and Leu[1063] in the BL2. The combined substitution of Pro[955] and Leu[1063] to Ala (P955A:L1063A) decreased the FRIP8 FRET signal, although single amino acid substitutions with each residue separately had no effect (Fig. 4h and Supplementary Fig. 4b). Importantly, despite mutations in the ubiquitin-binding pocket, P955A:L1063A enhanced the deubiquitinating activity of full-length USP8 (Fig. 4i and Supplementary Fig. 4e). Taken together with the effects of F1014A (Fig. 4b, c), these results clearly demonstrate a binding–inhibition relationship. Notably, Pro[955] and Leu[1063] as well as Phe[1014] are highly conserved among vertebrates (Supplementary Fig. 4f).

**14-3-3 inhibits USP8 enzyme activity partly by enhancing the interaction between the WW-like and USP domains.** Previous studies have shown that 14-3-3 binds to USP8 in a phosphorylation-dependent manner[29]. We measured the levels of phosphorylation in the 14-3-3-binding motif in total cell lysates and in absorbed fractions with GST-tagged 14-3-3. In the latter, all USP8 molecules should be phosphorylated. By comparing the results, we estimated that ~60% of cellular USP8 is phosphorylated at the 14-3-3-binding motif (Supplementary Fig. 5a).

R18 is a peptide that inhibits the interaction of 14-3-3 with its ligands[37]. Consistent with previous reports, R18 treatment induced the dissociation of 14-3-3 from USP8 (Fig. 5a) and enhanced the deubiquitinating activity of USP8 immunoprecipitates (Fig. 5b and Supplementary Fig. 5b). FRIP8 harbours a 14-3-3-binding motif between the WW-like and USP domains (Fig. 3d). Importantly, R18 treatment induced the dissociation of 14-3-3 from FRIP8 immunoprecipitates (Fig. 5c) and reduced the FRET signal (Fig. 5d). These results indicate that 14-3-3 enhances the interaction between the WW-like and USP domains.

The deletion of Ser[718] (ΔS718) is a common USP8 mutation in Cushing's disease that causes dissociation from the 14-3-3 protein[31,38]. The loss of polar contact between the phosphate group at Ser718 and a basic pocket of 14-3-3 can explain the dissociation[30]. A previous report indicated that the Ser[718] deletion enhances the deubiquitinating activity of USP8[31]. We compared the effects of Ser[718] or the WW-like domain deletion on the deubiquitinating activity of USP8. The deletion of Ser[718] enhanced this activity more weakly than the deletion of the WW-like domain (Fig. 5e and Supplementary Fig. 5c). Importantly, Ser[718] deletion was found to induce the dissociation of 14-3-3 from FRIP8 immunoprecipitates (Fig. 5f) and to slightly reduce the FRET signal (Fig. 5g and Supplementary Fig. 5d). It indicates that Ser[718] deletion significantly, but incompletely, inhibited the interaction between the WW-like and USP domains. Collectively, Ser[718] deletion enhanced the deubiquitinating activity of USP8, partly by suppressing interactions between the WW-like and USP domains.

**Discussion**

In this study, we showed that the USP8 WW-like domain and ubiquitin competitively bind to the ubiquitin-binding pocket of USP8. In most WW domains, a few aromatic or hydrophobic side chains on the intrinsic β-sheet make contact with short hydrophobic sequences (e.g. Pro-Pro-X-Tyr) in ligand proteins[34,39]. In contrast, our docking simulation highlights a distinct feature of the USP8 WW-like domain: several amino acids on the β-sheet and at the peripheral loops make contact with a large surface constructed by the different subdomains of the USP domain.

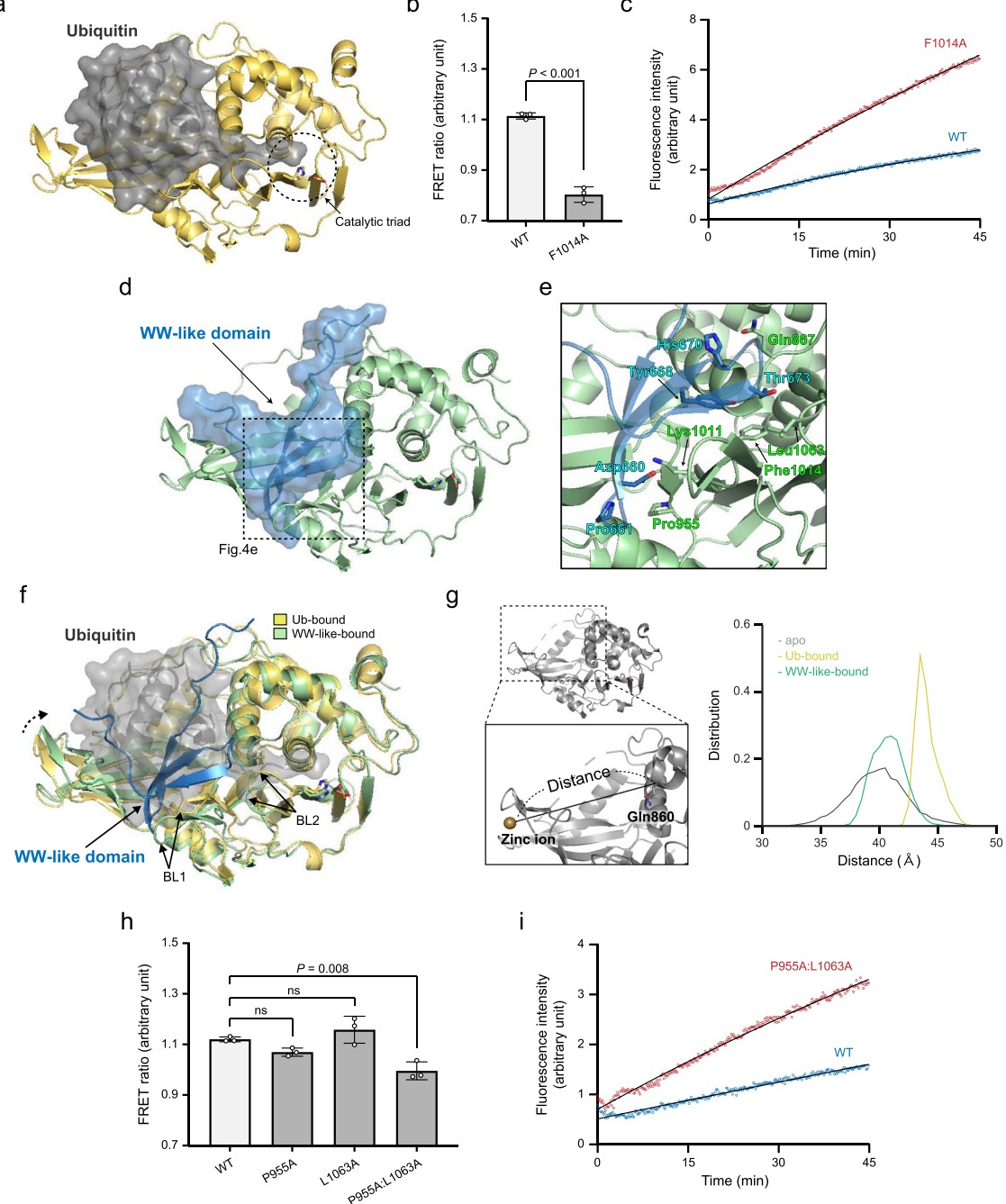

**Fig. 4 Structural modelling of the WW-like and USP domain complex. a** Model structure of the USP8 USP domain in ubiquitin-bound form. The USP domain and the bound ubiquitin are shown in yellow and grey, respectively. **b** The FRET ratio of FRIP8[F1014A], analysed using similar methods to those described in Fig. 3g. The graph shows the means ± standard deviations of three independent experiments. Statistical significance was determined using Student's *t*-test. **c** Deubiquitination activities of USP8[F1014A] towards ubiquitin-AMC, analysed using similar methods to those described in Fig. 2e. **d** Complex structures of the WW-like domain (blue) and USP domain (green), as predicted by docking simulations using the structural models of the WW-like domain presented in Fig. 2c and that of the USP domain presented in Fig. 4a. **e** Plugging of the catalytic cleft by the WW-like domain in the model. **f** Narrowing of the entrance to the ubiquitin-binding pocket due to the interaction in the model. The USP domain in the ubiquitin-bound form and that in the WW-like domain-bound form are shown in yellow and green, respectively. **g** Distributions of the distance between zinc ions in the fingers subdomain and Cα atoms of Gln[860] during the MD simulations of the WW-like domain-bound (green), ubiquitin-bound (yellow) and apo (grey) forms. **h** FRET ratio of FRIP8[P955A], FRIP8[L1063A] or FRIP8[P955A:L1063A], analysed using similar methods to those described in Fig. 3g. The graph shows the means ± standard deviations of three independent experiments. Statistical significance against the WT was determined using one-way ANOVA and Tukey's post hoc test. ns, not significant. **i** Deubiquitination activities of USP8[P955A:L1063A] towards ubiquitin-AMC, analysed using similar methods to those described in Fig. 2e. See Supplementary Fig. 4.

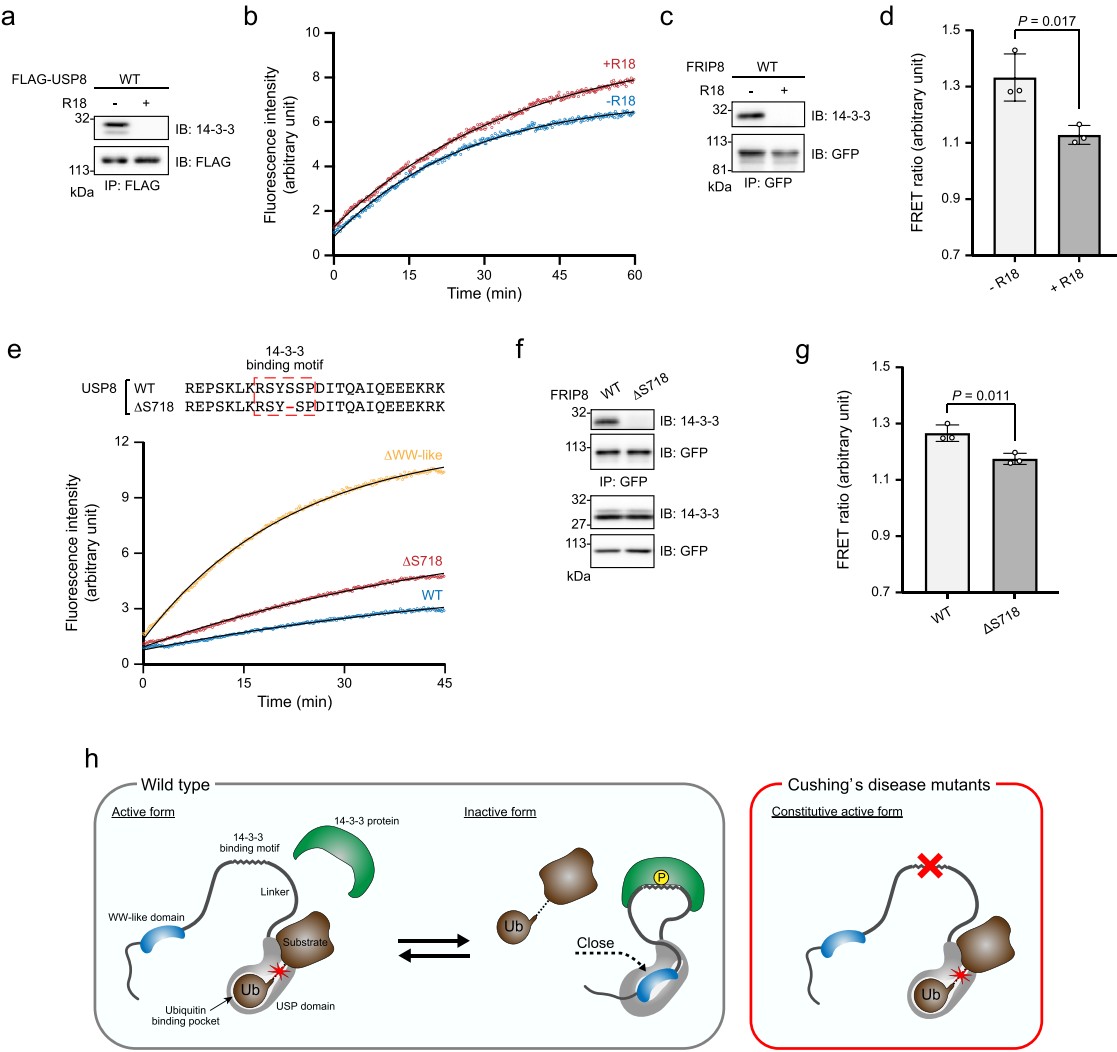

**Fig. 5 Effects of 14-3-3 on the interaction between the WW-like and USP domains. a** Effects of R18 on co-immunoprecipitation of USP8 with 14-3-3. HEK293 cells expressing FLAG-tagged USP8 were lysed. The lysate was incubated with 100 μM R18 before being subjected to immunoprecipitation and immunoblotting. **b** Effects of R18 on USP8 deubiquitinating activity. The lysates of HEK293 cells expressing FLAG-tagged USP8 were incubated with 100 μM R18 before they were subjected to an in vitro deubiquitination assay similar to that described in Fig. 2e. **c, d** Effects of R18 on FRIP8. HEK293 cells expressing FRIP8 and 14-3-3ε were lysed. The lysate was incubated with 100 μM R18 and then subjected to immunoprecipitation and subsequently immunoblotting (**c**). Samples were analysed using similar methods to those described in Fig. 3g (**d**). The graph shows the means ± standard deviations of three independent experiments. Statistical significance was determined using Student's t-test. **e** Deubiquitination activities of USP8s with Cushing's disease-associated mutations. The amino acid sequences of the mutants are shown in the top panel. The lysates of HEK293 cells expressing USP8 mutants were subjected to immunoprecipitation; the deubiquitinating activities in immunoprecipitates were then analysed using methods similar to those described in Fig. 2e. **f, g** Effects of Cushing's disease-associated mutations on FRIP8. HEK293 cells expressing FRIP8 mutants were lysed and subjected to immunoprecipitation. Samples were then analysed using immunoblotting (**f**) and a FRET assay (**g**). The graph shows the means ± standard deviations of three independent experiments. Statistical significance was determined using Student's t-test. **h** Working hypothesis of the autoinhibitory mechanism of USP8 and its impairment in Cushing's disease mutants.

Similarly, ubiquitin binds to the USP domains through a large contact area[40]. Through this large contact, the WW-like domain appears to plug the catalytic cleft of the USP domain. In addition, the WW-like domain alters the position of the fingers subdomain, probably by contacting Pro[955] in this subdomain, and thereby narrows the entrance to the pocket. To our knowledge, USP8 is the only USP family protein with a WW-like domain, which suggests that regulation via the WW-like domain is unique to USP8.

Several USPs are reported to be regulated by the intrinsic regions that physically interact with their catalytic domains[26,28,41–43]. Among these, USP25 had previously been the only USP for which the activity was inhibited on a well-defined structural basis. The

present study provides another example of the autoinhibition of USPs by an intrinsic region. In USP25, the autoinhibitory region occupies a space between the α5 helix and BL2[42,43]. Interestingly, the autoinhibitory regions of USP8 and USP25 similarly plug an entrance to the catalytic cleft in which the ubiquitin C-terminal tail should be embedded. In a case where a USP was inhibited by a compound, a similar space in USP14 was occupied by the small molecule inhibitor IU1[44]. Therefore, this space could potentially be utilised for the effective inhibition of various USPs by intrinsic regulatory motifs and chemical inhibitors.

We found that most cellular USP8s are bound to 14-3-3. Our analyses also indicated that 14-3-3 inhibits USP8's deubiquitinating activity by enhancing the interaction between the WW-like and

USP domains. Because the 14-3-3-binding motif is located at an unstructured region between the WW-like and USP domains, one possibility is that binding shortens the distance between these domains and thereby enhances their complex formation (Fig. 5h). However, further studies are warranted to elucidate the underlying mechanism.

The 14-3-3-binding motif is a hot spot at which somatic mutations frequently occur in Cushing's disease[31]. This study revealed a possible part of the pathogenic mechanism underlying Cushing's disease: the inability of USP8 to bind to 14-3-3 decreases the interaction between the WW-like and USP domains, which causes USP8 hyperactivation (Fig. 5h). To date, studies on Cushing's disease tumours have found various mutations around the 14-3-3-binding motif but none in the WW-like domain. Mutations in the WW-like domain may cause cell death rather than ACTH hypersecretion by enhancing USP8 deubiquitinating activity too strongly or by affecting the other functions of USP8. A previous study showed that mutations within the 14-3-3-binding motif result in the cleavage of USP8, thereby producing a 40-kDa C-terminal fragment (C40) with excess deubiquitination activity[31]. Surprisingly, we did not detect C40 in our study. Because different cell lines (Cos-7 cells vs. HEK293/HEK293T cells) were used in the two studies, C40 may be produced in a cell type-specific manner. It will, therefore, be important to determine whether significant levels of C40 are produced in the corticotrophs of patients with Cushing's disease. Nevertheless, our study demonstrated that Cushing's disease-associated mutations release USP8 autoinhibition, which causes high enzymatic activity even without its cleavage.

In conclusion, we found that the USP8 WW-like domain functions as an autoinhibitory region by binding to the ubiquitin-binding pocket of the catalytic domain. In addition, we found that 14-3-3 binds to USP8 and thus enhances the autoinhibitory interaction. In cells, USP8 deubiquitinating activity might be tuned by the association and dissociation of 14-3-3. In Cushing's disease, this regulation is apparently impaired by disease-associated mutations at the 14-3-3-binding motif. To build upon our findings, further studies will be required to determine the physiological and pathological significance of this regulatory mechanism of USP8.

## Methods

**cDNA preparation and plasmid construction.** Human USP8 cDNA[31], human 14-3-3ε cDNA[29], mouse STAM1 cDNA[45] and mouse STAM2 cDNA[46] were prepared elsewhere. Human Parkin and ubiquitin cDNAs were kindly donated by Drs Koji Yamano and Toshiaki Suzuki, respectively (Tokyo Metropolitan Institute of Medical Science, Japan). For the construction of expression plasmids, these cDNAs were subcloned into the following vectors: pME[31] for the expression of C-terminal FLAG-tagged USP8; pCDH-CMV-MCS-EF1-Puro (System Biosciences) for the viral infection and expression of N-terminal FLAG-tagged USP8; pcDNA3 (Invitrogen) for N-terminal HA-tagged ubiquitin; pFLAG-CMV2 (Sigma Aldrich) for N-terminal FLAG-tagged USP8, Parkin, STAM1 and STAM2; pmyc-CMV5 (gifted by Dr Jun Nakae, Keio University, Tokyo, Japan) for N-terminal myc-tagged USP8 and 14-3-3ε and pGEX-6P2 (GE Healthcare) for the bacterial expression of N-terminal GST-tagged USP domain (aa 756–1118).

For the construction of single-molecule FRET probes, human USP8 (aa 645–684) was subcloned into pEYFP-N1 (Clonetech). ECFP with flexible spacers comprising a ten-aa peptide (-Gly-Gly-Ser-Ala-Gly-Gly-Ser-Ala-Gly-Gly-) was amplified through PCR using pECFP-N1 (Clonetech)[47,48]. For ECFP insertion, the restriction sites for EcoRI and BamHI were inserted in the following locations of the USP domain: gaps between aa 816 and 817 as insert-1, aa 898 and 899 as insert-2 (FRIP8), aa 959 and 960 as insert-3, aa 1045 and 1046 as insert-4 and aa 1074 and 1075 as insert-5[19]. ECFP with the spacers was subcloned in a stepwise manner into these EcoRI and BamHI sites. For the construction of His-tagged FRIP8, His-tag sequence was fused to the C-terminal of FRIP8. Site-specific mutagenesis was achieved using PCR with a Prime STAR Mutagenesis Basal Kit (Takara).

**Protein preparation.** A GST-tagged USP domain (aa 756–1118) was constructed in *Escherichia coli* Rosetta. Protein expression was induced by the stimulation of cells with 0.5 mM IPTG, which was followed by overnight incubation at 15 °C. Cells were then lysed in ice-cold PBS supplemented with 1% Triton X-100 and

protease inhibitors (1 μg/ml each of aprotinin, leupeptin and pepstatin A). After centrifugation, the supernatants were mixed with glutathione sepharose 4B (GE Healthcare). After 1-h incubation at 4 °C, beads were washed five times with PBS and then incubated in ice-cold elution buffer A (50 mM Tris-HCl, pH 8.0; 1 mM EDTA; 1 mM dithiothreitol (DTT) and 10 mM reduced glutathione; Wako) at 4 °C for 30 min. The purity and concentration of the eluates were validated using SDS-PAGE and CBB staining using Quick CBB (Wako).

His-tagged FRIP8 was expressed in HEK293T cells. Cell culture and transfection are described below. Cells were lysed in ice-cold lysis buffer (50 mM Tris-HCl, pH 7.4; 150 mM NaCl; 50 mM NaF and 1% Triton X-100) supplemented with 10 mM imidazole and protease inhibitors. After centrifugation, the supernatants were mixed with Ni-NTA agarose beads (Qiagen). After 2-h incubation at 4 °C, the beads were washed five times with Tris-buffered saline (TBS; 20 mM Tris-HCl, pH 7.4 and 150 mM NaCl) and then incubated in ice-cold elution buffer B (20 mM Tris-HCl, pH 7.4; 150 mM NaCl and 250 mM imidazole) at 4 °C for 30 min. The purity was validated using immunoblotting.

For the in vitro deubiquitination assay and ubiquitin-VME labelling, FLAG-tagged USP8 (wild-type and mutants) were expressed in HEK293 cells (for ubiquitin-AMC assay) or HEK293T cells (for ubiquitin chain cleavage assay and ubiquitin-VME labelling). Cell lysis and immunoprecipitation were performed as described below. USP8 levels in each sample were compared using immunoblotting, with an equal number of USP8 molecules being used in the assay.

**Cell culture, plasmid transfection and lentivirus infection.** HEK293, HEK293T and HeLa cells were grown in Dulbecco's modified Eagle's medium (Nacalai Tesque) supplemented with 10% foetal bovine serum (FBS), 100 units/ml of penicillin and 0.1 mg/ml of streptomycin at 37 °C and 5% $CO_2$. To stimulate cells with EGF, they were cultured in the presence of 0.5% FBS for 24 h and then incubated with 100 ng/ml human EGF (PeproTech) at 37 °C for 5 min.

Plasmid transfections were performed using polyethyleneimine (Polyscience) according to the standard protocol. Cells were subjected to analyses 24–48 h after transfection.

For lentivirus infection, HEK293T cells were transfected with pCDH-CMV-FLAG-USP8-EF1-Puro, psPAX2 (Addgene #12260) and pCMV-VSV-G (Addgene #8454). After 2 days from the transfection, the medium containing the virus was collected and filtrated. HeLa cells were infected with the virus medium in the presence of 8 μg/ml polybrene (Nacalai Tesque). After 2 days from the infection, the cells were cultured in the presence of 0.8 μg/ml puromycin. Surviving cells were used for experiments.

**Cell lysis, immunoprecipitation and immunoblotting.** Cells were lysed with ice-cold lysis buffer (50 mM Tris-HCl, pH 7.4; 150 mM NaCl; 50 mM NaF and 1% Triton X-100) supplemented with protein inhibitors. To detect ubiquitin signals by immunoblotting, 2 mM N-ethylmaleimide (Sigma Aldrich) was added. For the immunoprecipitation of FLAG-tagged USP8 for the deubiquitination assay, 1 mM DTT (Nacalai Tesque) was added. After the centrifugation of the lysates, the supernatants were collected. To examine the effects of R18 peptide (Sigma Aldrich) on the interaction of 14-3-3 and USP8 or FRIP8, the lysates were incubated with 100 μM R18 peptide at 4 °C for 30 min.

Immunoprecipitation was performed following the manufacturer's recommended procedures. Anti-FLAG M2 antibody-conjugated agarose beads (Sigma Aldrich), anti-EGFR antibody (#MI-12-1, MBL, 1 μg), anti-GFP antibody (#M-048-3, MBL, 1 μg) and Protein A Sepharose (GE Healthcare) were used for immunoprecipitation. After immunoprecipitation, the beads were washed five times with lysis buffer. To elute FLAG-tagged proteins, the beads were incubated in TBS with 200 ng/μl FLAG peptide (Sigma Aldrich) at 4 °C for 30 min.

Samples were incubated in SDS-PAGE sample buffer (62.5 mM Tris-HCl, pH 6.8; 2% SDS; 5% 2-mercapto ethanol; 10% glycerol and 0.1 mg/ml bromophenol blue) at 98 °C for 5 min. To detect the ubiquitin signals, samples were incubated in NuPAGE LDS sample buffer with a reducing agent (Thermo Scientific) at 37 °C for 20 min. Samples were separated using SDS-PAGE or NuPAGE (Thermo Scientific).

Immunoblotting was performed following standard procedures. The primary antibodies used for immunoblotting were as follows: anti-FLAG (clone 1E6, Wako, 1:1000 dilution), anti-ubiquitin (#D058-3, MBL, 1:1000; #3936, Cell Signalling, 2 μg/ml), anti-EGFR (#MI-12-1, MBL, 1:2000), anti-USP8 (1:200)[49], anti-HRS (1:100)[50], anti-STAM1 (1:200)[51], anti-α-tubulin (clone 10G10, Wako, 1:2000), anti-HA (clone 3F10, Sigma Aldrich, 100 ng/μl; #SC-53516, Santa Cruz, 1 μg/ml), anti-Myc (#SC-40, Santa Cruz, 1:400; clone 9E10, hybridoma supernatant, 1:5), anti-GFP (#A-11122, Thermo Scientific, 1:2000) and anti-14-3-3 (#SC-1657, Santa Cruz, 1:1000) antibodies. The secondary antibodies used were as follows: peroxidase-conjugated anti-mouse, anti-rat and anti-rabbit IgG antibodies (GE Healthcare). Blots were detected using ECL Prime Western Blotting Detection Reagents (GE Healthcare) and ImageQuant LAS 4000 Mini (GE Healthcare).

**In vitro deubiquitination assay.** For ubiquitin chain cleavage assay, USP8 (wild-type or the mutants) was incubated with 25 ng/μl Lys63-linked ubiquitin tetramers (Boston Biochem) in TBS with 10 mM DTT at 37 °C for the indicated times in the presence or absence of R18 peptide. Samples were subjected to SDS-PAGE or NuPAGE and then to gel staining with a Silver Stain MS Kit (Wako).

For ubiquitin-AMC assay, USP8 was mixed with 1 μM ubiquitin-AMC (Boston Biochem or Life Sensors) and 10 mM DTT in TBS in 96-well medium-binding, flat-bottom, black plates (Thermo Scientific). To examine the effects of R18 peptide, USP8 was incubated with 100 μM R18 peptide at 4 °C for 30 min before the assay. Plates were set on Varioskan LUX (Thermo Scientific) and incubated at 37 °C. During the incubation, the fluorescence intensity of AMC released from ubiquitin was measured at 10-s intervals using 345 and 445 nm as the excitation and emission wavelengths, respectively. The fluorescence intensity of the sample without USP8 was measured as the background and then subtracted from each value. Non-linear regression curves were plotted using the equation for one-phase association using Prism 9 (GraphPad Software).

**Ubiquitin-VME labelling**. USP8 immunoprecipitates or His-tagged FRIP8 precipitates were incubated with 1 μM ubiquitin-VME (Boston Biochem) in TBS supplemented with 1 mM DTT at 37 °C for 5 or 30 min, respectively. Samples were then subjected to immunoblotting and a FRET assay.

**Immunofluorescence**. HeLa cells on coverslips were fixed with PBS containing 4% paraformaldehyde, permeabilised with PBS containing 0.2% Triton X-100 and blocked with PBS containing 5% FBS. Cells were stained with mouse anti-FLAG antibody (clone M2, Sigma Aldrich, 1 μg/ml) and Alexa Fluor 488-conjugated anti-mouse IgG secondary antibody (Invitrogen) based on standard procedures. Coverslips were mounted on slides with Fluoroshield mounting medium (Immuno-BioScience). Fluorescence images were captured with a laser-scanning confocal microscope (LSM 780, Carl Zeiss).

**Amino acid sequence alignment**. Protein BLAST (https://blast.ncbi.nlm.nih.gov/) was used to identify protein sequences similar to USP8 aa 645–684. Sequence similarity was analysed using ClustalW (https://clustalw.ddbj.nig.ac.jp/); the colours in the alignment followed the ClustalX scheme. Amino acid sequences of human MAGI1 (accession code: Q96QZ7), human SAV1 (Q9H4B6), human WWOX (Q9NZC7) and vertebrate USP8s were based on annotated information present in the UniProt database (https://www.uniprot.org/).

**In silico structural modelling**. In silico structural modelling of the WW-like and USP domains and their complex was performed. In brief, (1) homology modelling of the WW-like domain was first performed using the MAGI1 WW (PDB ID: 2YSE), (2) the structural model of the USP domain in the ubiquitin-bound form was then constructed using the ubiquitin-bound USP2 USP domain (PDB ID: 2HD5) as the template for homology modelling and (3) the protein–protein docking simulations of the two domains were executed using ClusPro (https://cluspro.org/). The MD simulations of the derived modelled structures were also performed to examine their stabilities in the physiological condition; the details are described below.

Using data on the sequence similarity with the MAGI1 WW domain (Fig. 2b) and structural data from an NMR experiment (PDB ID: 2YSE), the structure of the WW-like domain in USP8 (aa 645–684) was modelled using MODELLER[52]. To perform MD simulations, a rectangular simulation box was constructed with a margin of 12 Å to its boundary, ensuring the following dimensions: 58 Å × 51 Å × 58 Å. The solution system contained 4139 TIP3P water molecules[53] together with one sodium ion to neutralise the simulation system, resulting in 13,045 atoms in total. AMBER ff14SB[54] was used for the potential energy of the all-atom protein. For the zinc ion in the fingers subdomain, a Zinc AMBER force field[55,56] was used. The MD simulations were performed using AMBER 16[57] under constant temperature and pressure (NPT) conditions of $T = 300$ K and $P = 1$ atm, using the Berendsen thermostat and barostat[58], with a relaxation time of 1 ps, and using the particle mesh Ewald method[59] for electrostatic interactions. The simulation length was 1 μs, together with a 2-fs time step using constraining bonds involving hydrogen atoms via the SHAKE algorithm[60]. The simulation of the W655S mutant was also performed as described above, after replacing the wild-type W655 with Ser using MODELLER[52].

A structural model of the USP domain (aa 756–1110) was also constructed to investigate potential interactions with the WW-like domain. Because the open form with bound ubiquitin was not solved for USP8, the ubiquitin-bound USP2 USP domain (PDB ID: 2HD5) was used as a template for homology modelling. The modelled structure of the USP8 USP domain with a 76-residue ubiquitin bound to it was then solvated in a simulation box with the dimensions 103 Å × 84 Å × 91 Å (including 65,415 atoms in total) before being simulated for 1 μs, as described above. The stability of the modelled ubiquitin-USP domain complex was observed by the small root mean square distribution on the Cα atom resolution (Cα-RMSD) of the overall structure being < 2.6 Å from that of the initial simulation model. The structure of the USP domain in the last simulation trajectory time step was then used for the subsequent simulation of docking with the WW-like domain. The MD simulation of the apo USP domain in the closed-form was also performed using the crystal structure (PDB ID: 2GFO) as the initial model.

The structural basis of stable complex formation between the USP domain and the WW-like domain was determined using the docking simulation. A protein–protein docking simulation was performed using the protein–protein interaction server ClusPro together with standard parameter sets[61] to generate putative interaction complexes. We chose six model candidates from these complexes using the criterion that the F1014 of the USP domain existed in the interface with the WW-like domain. We then performed a 100-ns MD simulation for each model candidate, as described above. Using the associated MD trajectories, the number of atom contacts, $N_c$, was calculated as the number of non-hydrogen atom pairs with $r < 4$ Å, and the binding free energy between the USP and WW-like domains, $\Delta G_{bind}$, was evaluated using the MM-GB/SA module of AMBER 16[57]. The structure with the largest $N_c$ and the lowest $\Delta G_{bind}$ was selected as the most probable complex model (Supplementary Fig. 4d). An additional 1-μs MD simulation of the resulting USP/WW-like domain complex showed that the overall Cα-RMSD was < 3.5 Å from that of the initial simulation model, indicating that the derived complex structure had substantial stability under physiological conditions.

**Pull-down assay**. HEK293T cells expressing indicated prey proteins were lysed with lysis buffer supplemented with protein inhibitors. The lysates were incubated with 5 μM GST or GST-tagged USP domain at 4 °C for 2 h and then incubated in the presence of glutathione sepharose 4B at 4 °C for 90 min. The beads were then washed five times with lysis buffer and samples were subjected to immunoblotting.

**FRET assay**. Samples were added to 96-well medium-binding, flat-bottom, black plates. Fluorescence intensities were measured using Varioskan LUX or F2700 (Hitachi Hightech) at 5- or 0.5-nm wavelength intervals, respectively, with 433 nm as the excitation wavelength and 460–570 nm as the emission wavelength range. Negative control samples were incubated with 10 ng/μl trypsin (Thermo Scientific) at 4 °C for 60 min, and then fluorescence intensities were measured. To examine the effects of R18 peptide, samples were incubated with 100 μM R18 peptide at 4 °C for 30 min before the assay. In blank wells, assay buffer or lysates derived from cells that did not express FRET probes were also prepared. These fluorescence intensities were measured as background signals and subtracted from the measured values. The FRET ratio was calculated based on the peak intensity at the EYFP emission wavelength divided by that at the ECFP emission wavelength.

**Statistical analysis**. Data were presented as means ± standard deviations. Data were analysed using two-tailed Student's t-tests or one-way factorial ANOVA followed by Tukey–Kramer multiple comparison post hoc tests. P values of <0.05 were considered significant.

**Un-cropped scan data**. Un-cropped scan data of blots and staining are shown in Supplementary Figs. 6 and 7.

**Reporting Summary**. Further information on research design is available in the Nature Research Reporting Summary linked to this article.

## Data availability

Data supporting the study findings are available from the corresponding authors upon reasonable request and have been included as Supplementary Data 1 with this publication.

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

## Acknowledgements

We thank Dr. Sayaka Yasuda (Tokyo Metropolitan Institute of Medical Science) for helpful discussions and Dr. Koji Yamano (Tokyo Metropolitan Institute of Medical Science) for the kind donation of Parkin plasmids. We also acknowledge the help of the Biomaterials Analysis Division (Tokyo Institute of Technology) with DNA sequencing analysis. This work was supported by JSPS KAKENHI Grant Number 17K08625, 19H05289 and 21H00276 to T.F.; JSPS KAKENHI Grant Number 15H04293 to M.Komada; Platform Project for Supporting Drug Discovery and Life Science Research (Basis for Supporting Innovative Drug Discovery and Life Science Research) from AMED under Grant Number JP21am0101109 to M.I. and the Nagase Science and Technology Foundation.

## Author contributions

M.Komada and T.F. conceived and supervised the study; K.K., K.A., K.M., T.K., A.E. and T.F. designed the experiments; K.K., K.A., K.M., M.Kubo and A.E. performed the

experiments and analysed data; K.K. and T.F. drafted the manuscript; T.K., A.Kidera, M.I., A.Kato and M.Komada critically revised the manuscript.

## Competing interests

The authors declare no competing interests.
