## [Peer Review File · Communications Biology]

REVIEWERS' COMMENTS:

Reviewer #1 (Remarks to the Author):

The manuscript by Kakiyama et al aims to examine the molecular basis of USP8 auto-inhibition by its WW-like domain. The manuscript present three main findings: 1) amino acids 645–684 of USP8 as an autoinhibitory region; 2) amino acids 645–684 form WW-like domain, which plugs the catalytic cleft, and narrows the entrance to the ubiquitin domain structure; 3) 14-3-3 was found to inhibit USP8 enzyme activity partly by enhancing the interaction between the WW-like and USP domains. The study reported a mechanism underlying USP8 auto-inhibition, and provided new insights into USP8 mutation-caused Cushing's diseases. This is interesting paper and the experimental results have supported the main conclusions. However, there are a lot of grammar errors needed to be corrected by naïve English speaker before consideration of publications.

Reviewer #2 (Remarks to the Author):

The manuscript "The molecular basis of ubiquitin-specific protease 8 autoinhibition by the WW-like domain" by Kakiyama et al is an elegant study to decipher an important regulatory mechanism of the Cushing's disease-related de-ubiquitinase USP8. Mutations in this enzyme is the most prominent genetic alteration in Cushing's and consequently to understand the mechanistic basis of its regulation is of utmost relevance.

The authors showed in a number of well-designed experiments how the WW-like domain in USP8 is involved in the regulation of the enzymatic activity and how this regulation is connected to Cushing's disease mutations and USP8's regulation by 14-3-3 proteins.

A comment on the 14-3-3 axis of regulation of USP8 is also the only point that I would suggest to improve the manuscript by discussion or at least mentioning the crystallographic and biophysical study of the 14-3-3 binding motif in USP8 (Centorrino et al., FEBS Lett. 2018 Apr;592(7):1211-1220). This study showed both the structure of the interface of the USP8/14-3-3 complex as well as the influence of mutations found in Cushing's disease on this interaction and is of relevance to this manuscript.

Our answer to reviewers' comments:

The comment of reviewer #1:

The manuscript by Kakihara et al aims to examine the molecular basis of USP8 auto-inhibition by its WW-like domain. The manuscript present three main findings: 1) amino acids 645–684 of USP8 as an autoinhibitory region; 2) amino acids 645–684 form WW-like domain, which plugs the catalytic cleft, and narrows the entrance to the ubiquitin domain structure; 3) 14-3-3 was found to inhibit USP8 enzyme activity partly by enhancing the interaction between the WW-like and USP domains. The study reported a mechanism underlying USP8 auto-inhibition, and provided new insights into USP8 mutation-caused Cushing' diseases. This is interesting paper and the experimental results have supported the main conclusions. However, there are a lot of grammar errors needed to be corrected by naïve English speaker before consideration of publications.

Thank you for your appreciation of our research. We had a native English speaker correct our manuscript, including the title, abstract and the main text. A certificate of proofreading is also attached below.

CERTIFICATE OF EDITING

This is to certify that the paper titled Molecular basis of ubiquitin-specific protease 8 autoinhibition by the WW-like domain commissioned to us by Toshiaki Fukushima has been edited for English language, grammar, punctuation, and spelling by Enago, the editing brand of Crimson Interactive Pvt. Ltd under Normal Editing B2C.

Issued Enago, Crimson Interactive Pvt. Ltd.
1001, Techniplex- II, S. V. Road,
Goregaon (W), Mumbai 400062, India.
Phone: 03-5050-5374
Fax: 03-4496-4934

Disclaimer: The intent of the author's message has been preserved during the editing process. The author is free to accept or reject our changes in the document after reviewing our edits. This certificate has been awarded at the time of sharing the final edited version (full file or sections of the file) with the author. Enago does not bear any responsibility for any alterations done by the author to the edited document post **16 Sep 2021**.

Japan www.enago.jp, www.ulatus.jp, www.voxtab.jp
Taiwan www.enago.tw, www.ulatus.tw
China www.enago.cn, www.ulatus.cn
Brazil www.enago.com.br, www.ulatus.com.br
Germany www.enago.de

Russia www.enago.ru
Arabic www.enago.ae
Turkey www.enago.com.tr
S. Korea www.enago.co.kr
Global www.enago.com, www.ulatus.com, www.voxtab.com

About Crimson:
Crimson Interactive pvt ltd is one of the world's leading academic research support services. Since 2005, we've supported over 2 million researchers in 125 countries with their publication goals.

The comment of reviewer #2:

The manuscript "The molecular basis of ubiquitin-specific protease 8 autoinhibition by the WW-like domain" by Kakihara et al is an elegant study to decipher an important regulatory mechanism of the Cushing's disease-related de-ubiquitinase USP8. Mutations in this enzyme is the most prominent genetic alteration in Cushing's and consequently to understand the mechanistic basis of its regulation is of utmost relevance.

The authors showed in a number of well-designed experiments how the WW-like domain in USP8 is involved in the regulation of the enzymatic activity and how this regulation is connected to Cushing's disease mutations and USP8's regulation by 14-3-3 proteins.

A comment on the 14-3-3 axis of regulation of USP8 is also the only point that I would suggest to improve the manuscript by discussion or at least mentioning the crystallographic and biophysical study of the 14-3-3 binding motif in USP8 (Centorrino et al., FEBS Lett. 2018 Apr;592(7):1211-1220). This study showed both the structure of the interface of the USP8/14-3-3 complex as well as the influence of mutations found in Cushing's disease on this interaction and is of relevance to this manuscript.

Thank you for your suggestion. We cited this important report and improved our manuscript as follows.

The 3rd paragraph in the introduction section:

“The binding of USP8, which has a phosphorylation-dependent 14-3-3-binding motif (Fig. 1b), to 14-3-3 protein inhibits its enzyme activity [29]. The binding mode exhibits the characteristic features of the mode I of 14-3-3 binding (RXX[pS/pT]XP) [30].”

The 3rd paragraph in the subsection “14-3-3 inhibits USP8 enzyme activity partly by enhancing the interaction between the WW-like and USP domains” in the result section:

“The deletion of Ser⁷¹⁸ (Δ S718) is a common USP8 mutation in Cushing’s disease that causes dissociation from the 14-3-3 protein [31, 38]. The loss of polar contact between the phosphate group at Ser718 and a basic pocket of 14-3-3 can explain the dissociation [30].”